# Geospatial patterns, and individual and community levels factors of cesarean section deliveries in Ethiopia: A spatial and multilevel analysis

**Abel Endawkie**[1]*, **Shimels Derso Kebede**[2], **Natnael Kebede**[3], **Mengistu Mera Mihiretu**[4], **Ermias Bekele Enyew**[2], **Kokeb Ayele**[3], **Lakew Asmare**[1], **Fekade Demeke Bayou**[1], **Mastewal Arefaynie**[5], **Yawkal Tsega**[4]

1 Department of Epidemiology and Biostatistics, School of Public Health, College of Medicine and Health Sciences, Wollo University, Dessie, Ethiopia, 2 Department of Health Informatics, School of Public Health, College of Medicine and Health Sciences, Wollo University, Dessie, Ethiopia, 3 Department of Health Promotion, School of Public Health College of Medicine Health Sciences, Wollo University, Dessie, Ethiopia, 4 Department of Health System and Management, School of Public Health, College of Medicine and Health Sciences, Wollo University, Dessie, Ethiopia, 5 Department of Reproductive and Family Health, School of Public Health, College of Medicine and Health Sciences, Wollo University, Dessie, Ethiopia

* abelendawkie@gmail.com

**Data Availability Statement:** Data are available in a public, open access repository. Data for this study were sourced from Demographic and Health

## Abstract

### Background

Cesarean Section (CS) is the most popular surgery worldwide in obstetric care to save a mother's or the fetus's life. The prevalence of CS delivery in Ethiopia was 0.7% and 1.9% in 2000 and 2016 respectively and its spatial distribution and variation in Ethiopia are limited. This study provides evidence for healthcare providers and pregnant women on the national CS geospatial distribution and variation to promote evidence-based decision-making and improve maternal and neonatal outcomes. Therefore, this study aimed to determine geospatial patterns and individual and community-level factors of CS deliveries in Ethiopia.

### Method

A secondary data analysis of 5,527 weighted samples of mothers using the 2019 Ethiopian mini demographic and health survey was conducted. The spatial hotspot analysis using Getis-Ord Gi* hot spot analysis of ArcGIS version 10.7.1 was used to show the spatial cluster of CS and multilevel mixed effect logistic regression analyses were employed. Statistical significance was declared at p-value <0.05 and adjusted odds ratio (AOR) with 95% confidence interval (CI) was reported.

### Result

The prevalence of CS delivery in Ethiopia was 5.4% with 95%CI (4.6%, 6.4%). The spatial autocorrelation shows CS was clustered in Ethiopia (global Moran's index = 1.009, and p-value<0.001). Spatial hotspot indicates CS was prevalent in Addis Ababa, Diredewa,

surveys (DHS) and available here: https://dhsprogram.com/data/available-datasets.cfm

**Funding:** The author(s) received no specific funding for this work.

**Competing interests:** The authors declare that they have no competing interests.

**Abbreviations:** AIC, Akakian information Criteria; AOR, Adjusted odds ratio; ANC, Antenatal care; BIC, Bayesian information Criteria; COR, Crud odds ratio; EA, Enumeration Area; ICC, Intra-Class Correlation; Mini-EDHS, Mini Demographic and Health Survey; MOR, Median Odds Ratio; PCV, Proportional Changes of Variance; SNNP, South Nation.

Oromo, and Somali regions. The odds of CS delivery were higher among mothers aged 24–35: AOR = 1.98, 95% CI (1.3, 3.1), and 35–49: AOR = 3.7, 95% CI(2.2, 6.1), live with female household heads: AOR = 1.9, 95% CI (1.2, 3.1), mothers with primary AOR = 1.6, 95% CI (1.07, 2.7), secondary: AOR = 2.4, 95% CI (1.3, 4.25) and higher education: AOR = 2.06, 95% CI (1.03, 4.2), multiple births: AOR = 8.1, 95% CI (3.4, 19.74), mothers in Addis Ababa: AOR = 3.4, 95% CI (1.1, 11.9) and Diredewa: AOR = 7, 95% CI (1.23, 40.7) as compared to their counterparts.

## Conclusion

In Ethiopia, CS deliveries remain below the World Health Organization estimate with distinct geospatial variation. Geographically, CS delivery is concentrated in urban areas like Addis Ababa and Diredewa, as well as in parts of the Somali and Oromia regions. Maternal age, female household head, Muslim religion, higher educational status, and multiple births at the individual level, and region at the community level were factors of CS delivery in Ethiopia. Therefore, efforts should be geared towards strategies and interventions to ensure fair access to CS delivery in line with WHO recommendations, especially in the regions where the CS delivery rate is below WHO estimates in Tigray, Amhara, Afar, and Benishangul Gumze regions.

## Introduction

Cesarean section (CS) is the most usually lifesaving surgical procedure in obstetric care. It is the birth of a fetus through a surgical incision made in the uterine and abdominal walls of a pregnant woman. Preventing maternal and newborn morbidity and mortality rates is the primary purpose of conducting it, either for fetal or maternal reasons [1]. Cesarean sections are more frequent than ever before, and this trend is expected to continue in the next ten years [2, 3], even though there is still a significant lack of safe CS in resource-constrained nations [4, 5].

Since 1985, the World Health Organization (WHO) has recommended that CS prevalence should be between 10–15% of all births among reproductive-age mothers in the country or specific community [6]. According to the WHO statement, a CS prevalence of less than 5% among reproductive-age mothers in the country or specific community indicates the unmet need, whereas a prevalence of CS greater than 15% among reproductive-age mothers might put the mother and new birth at risk [7].

The average CS prevalence according to recent statistics from 154 nations is 21.1% and it ranges from 8.2% in the least developed to 27.2% in the more developed countries [8]. In Africa CS delivery was 13% in Ghana [9], 2.1% in Nigeria [10], and 4.7% in Mozambique [11]. In Ethiopia CS delivery was 0.7% and 1.9% in 2000 and 2016 respectively [12, 13].

A worldwide population-based ecological study showed that CS rates greater than 10% are not linked to lower rates of maternal and newborn mortality [14]. The number of CS deliveries per 100 births, or the CS rate, has a significant linear connection with infant death, according to another ecological study [15]. Despite all of these facts, indicated CS has seen a large increase in practice in recent years [16]. Research conducted in Ethiopia showed that a rise in CS has been associated with both medically and non-medically indicated factors, such as the kind of health facility, socio-demographic traits, and health of mothers [17, 18]. A study conducted in Ethiopia indicates that CS delivery increased the risk of infection, a longer healing

period, significant bleeding, and an increase in iatrogenic injuries, breathing difficulties, and even death [19, 20]. Moreover, different studies reported different prevalence and associated factors for CS delivery in Ethiopia [3, 8, 21, 22].

Mother's age, birth order, first-parity births, Antenatal Care (ANC) visits, urban residence, higher education levels, and higher economic statuses were factors of CS delivery among mothers in Ethiopia [3, 8, 21–23]. These studies used small sample sizes which were not representative and were prone to selection bias [3, 8, 21, 22]. Moreover, in Ethiopia the spatial distribution and variation of CS delivery are limited. The study emphasizes the need to provide evidence for healthcare providers and pregnant women on the national CS delivery spatial distribution, variation, and individual and community level factors in using national representative data of the 2019 mini Ethiopian Demographic and Health Survey (Mini-EDHS) to promote evidence-based decision-making and improve maternal and neonatal outcomes. The 2019 Mini-EDHS is sufficient to provide nationwide data to analyze individual and community-level factors of CS delivery in Ethiopia. Therefore, this study aimed to determine geospatial patterns, and individual and community levels factors of cesarean section deliveries in Ethiopia using nationwide 2019 Mini-EDHS data.

## Method

### Study setting

The study was done in Ethiopia which is located in East Africa. It is bordered on the west by Sudan, on the east by Somalia and Djibouti, on the north by Eritrea, and the south by Kenya. The country has a total area of 1,112,000 square kilometers Ethiopia is divided into eleven regions and two municipal governments. Tigray, Afar, Amhara, Oromiya, Somali, Benishangul-Gumuz, Southern Nations Nationalities and People (SNNP), Gambela, Sidama, South Western, and Harari are among the regions involved. Addis Ababa and Dire Dawa are administrative cities. Data were accessed from their URL: www.dhsprogram.com by contacting them through personal accounts after justifying the reason for requesting the data [24].

### Study design and source population

The research was conducted using secondary data from a cross-sectional study used in the 2019 Mini-EDHS using children's records data. The source population was reproductive-age mothers (15–49) who gave birth before the survey.

### Sample size and sampling methods

A total of 5,526.93 ~ 5,527 weighted reproductive-age mothers were used in this analysis as a sample that was aimed to represent all of the country's regions and administrative cities from 5,753 mothers used in Mini-EDHS 2019. The sample was stratified and selected in two stages. In the first stage, stratification was conducted by region, and then each region was stratified as urban and rural. The sample size was then allocated using a probability-proportional allocation method. There were 305 enumeration areas (EAs) chosen for the 2019 Mini-EDHS (Fig 1).

### Variable measurement

Cesarean section delivery categorized as Yes"1" and No"0" was the dependent variable and the individual-level factors included: age, sex of household, religion, wealth index, marital status, educational status, ANC visit, types of birth, place of delivery, and community-level factors: region and residence were considered as the independent variable.

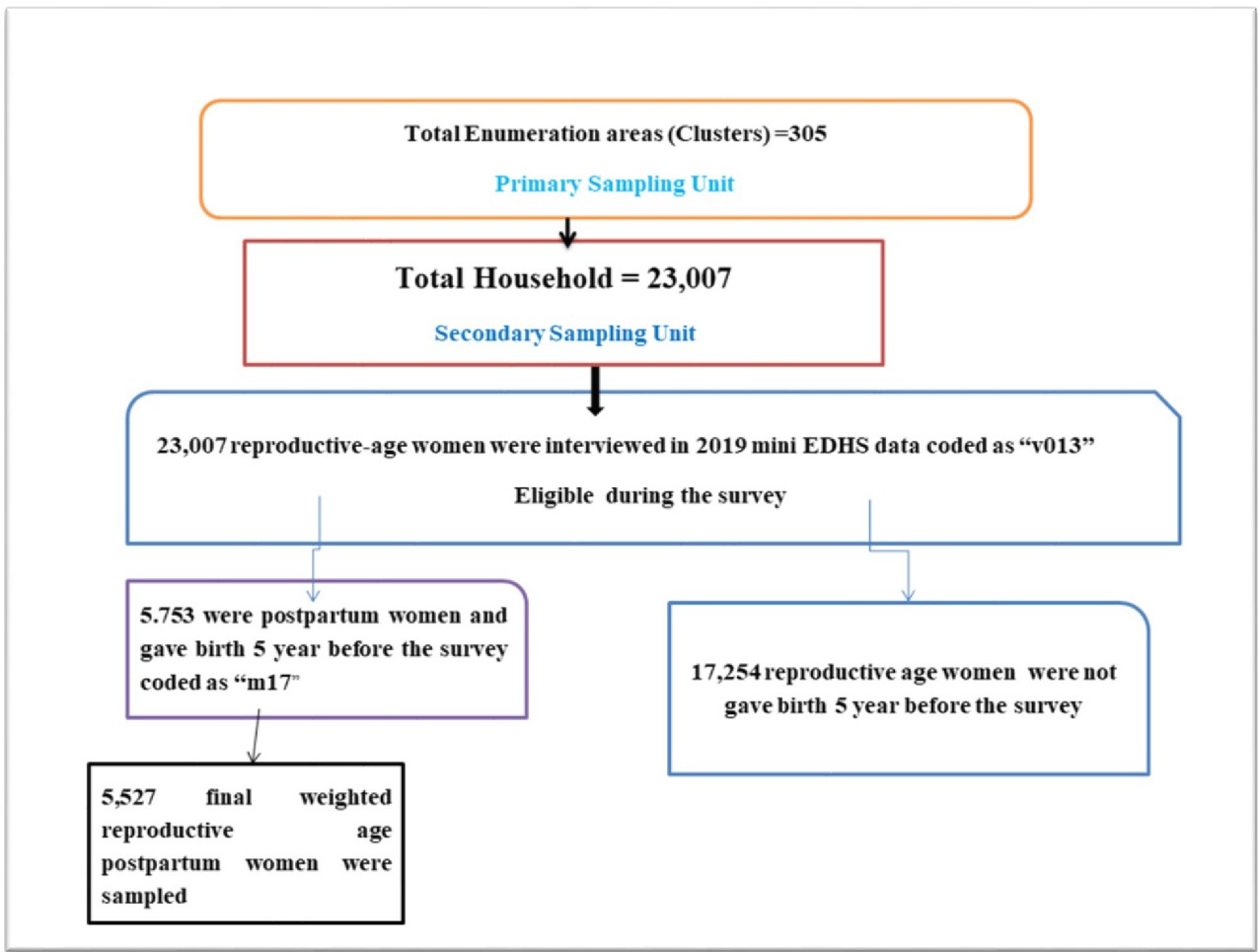

**Fig 1. Sampling and exclusion procedures to identify the final sample size of the study.**

### Data processing and analysis

Data used were extracted, cleaned, coded, and analyzed using STATA version 17 statistical software. Sample weights were done before further analysis to compensate for the unequal probability of selection between the strata that were geographically defined and for non-responses. Descriptive statistics were calculated and presented using tables, figures, and narratives.

### Spatial analysis

To execute the spatial analysis, the weighted proportions of outcome variables (CS delivery) with cluster number were computed in STATA version 17 and the spatial analysis was computed using QGIS version 3.28.15 and ArcGIS version 10.7.1 statistical software.

**Spatial distribution.** The descriptive spatial analysis was conducted to determine the spatial distribution of CS delivery in Ethiopia using QGIS version 3.28.15.

### Spatial autocorrelation analysis

The spatial autocorrelation (Global Moran's I) statistic measure is used to assess whether CS delivery is dispersed, clustered, or randomly distributed in Ethiopia. Moran's I value close to

− 1 shows dispersed CS delivery, close to + 1 shows clustered, and Moran's I value of zero shows randomly distributed [25]. In this study, spatial autocorrelation indicates CS delivery is clustered at a p-value of 0.0000 and Global Moran's I (MI) = 1.0009.

**Hotspot analysis.** The spatial hotspot analysis was conducted using Getis-Ord Gi* hot spot analysis of ArcGIS version 10.7.1 to examine where spatial clusters of CS delivery occur on the landscape by identifying areas with statistically significant clustering of relative high values (hot spots) or relative low values (cold spots). This measure indicates the presence or absence of significant spatial clustering, with the null hypothesis being that there is no difference in characteristics between a unit and its spatial neighbors [26–28].

**Spatial interpolation.** Spatial interpolation was used to estimate the proportion of CS delivery distribution in un-sampled areas, based on observed measurements in the sampled enumeration areas. Although there are different deterministic and geostatistical interpolation methods available, we opted to use the Ordinary Kriging spatial interpolation method since it incorporates spatial autocorrelation statistically optimizes the weight, and has low residual and mean square error [29].

**Multilevel-mixed-effect analysis.** Multilevel analysis was conducted after checking the eligibility of the data for multilevel analysis by using the intra-cluster correction coefficient (ICC). Since ICC is greater than 10% (ICC = 44.1%) calculated as below [30].

$$\text{ICC} = \frac{\delta 2}{\delta 2 + \pi 2/3} \tag{1}$$

Where δ2 indicates the estimated variance of clusters and π = 3.14

Multilevel analysis was considered appropriate due to the hierarchical nature of the DHS data as well as the estimation of individual and community-level effects. The log of the probability of a cesarean section was modeled using a two-level multilevel model as follows:

$$\text{Log}\left[\frac{\pi ij}{1 - \pi ij}\right] = \beta 0 + \beta 1 \text{Xij} + \text{B2 Zij} + \mu j + \text{eij}, \tag{2}$$

Where i and j are individual level and community level [31] units respectively; X and Z refer to individual and community level variables respectively; πij is the probability of cesarean section for the ith mothers in the jth community; β's indicates the fixed coefficients. (B0) is the intercept, the effect on the probability of cesarean section in the absence of influencing factors; and μj showed the random effect (the effect of the community on the cesarean section of the jth community) and eij showed random errors at the individual level.

By assuming each community had a different intercept (B0) and fixed coefficient (β), the clustered data nature and intra and inter-community variations were taken into account. Data organized into, analyses of cesarean section yes/no which has the binary response and runs mixed effect multi-level logistic regression model by considering two levels (individual and community levels). During analysis first, bi-variable multilevel logistic regression was fitted, and variables with p value less than 0.2 were selected to develop model I, model II, and the final model. The first model was, model-0 (empty model or null model), the second model was, model-I (analyzing only individual-level variables), the 3rd model was, model-II (analyzing only community-level variables), the last model, model-III (analyzing both community-level and individual-level variables based on the cut of the point). Variables with p-values less than 0.05 at model-III were significantly associated with cesarean section. The random effects (variations) were measured by using ICC (model-0), Median Odds Ratio (MOR) in (model-I and II), and Proportional Change in Variance (PCV) was measured to show variation between clusters. ICC shows the variation in cesarean section among mothers due to community

characteristics. MOR is the median value of the odds ratio between the area at the highest risk and the area at the lowest risk when randomly picking out two areas and it was calculated as follows.

$$\text{MOR} = \exp(\sqrt{2 + \delta2 + 0.6745}) \tag{3}$$

PCV measures the total variation attributed to individual-level variables and area-level variables in the final model (model-III). It is calculated as below.

$$\text{PCV} = \frac{\delta2 \text{ of null model} - \delta2 \text{ of each model}}{\delta2 \text{ of null model}} \tag{4}$$

Multicollinearity was checked among explanatory variables by using standard error at a cut of point±2 and there is no Multicollinearity since the standard errors were between±2. The appropriateness of the mixed model (model fitness) is checked using Inter-Class Correlation and model selection was based on Log Likelihood Ratio (LLR) or Akakian information criteria AIC and Bayesian information criteria (BIC).

**Ethical approval.**   No ethical approval was needed because we had used the demographic and health survey, which de-identifies all data before making it public, and the used mini DHS data sets are openly accessible. An authorization letter was requested to download the DHS data set and this was obtained from the Central Statistical Agency (CSA) after being requested at https://dhsprogram.com/. The dataset and all methods of this study were conducted according to the guidelines laid down in the Declaration of Helsinki principle and based on DHS research guidelines.

## Result

### Socio-demographic characteristics of mothers who gave birth before the survey

A total of, 5,527 weighted reproductive-age mothers [15–49] were included in the analysis. Among 205 reproductive-age mothers, 48% of mothers in the age group of 15–24 have delivered via cesarean section, from 2, 961 reproductive-age mothers, 2.9% of mothers who cannot read and write have delivered via cesarean section. Among 4160 reproductive-age mothers, 11.1% of mothers residing in rural areas had delivered via cesarean section (Table 1).

### Maternal health service utilization of mothers who gave birth before the survey

Regarding utilization of maternal health services utilization, 234 (8%) of mothers who had ANC visits delivered via cesarean section, and 270 (10.8) of mothers who delivered in private hospitals delivered via cesarean section (Table 2).

### Spatial distribution, hotspot, and interpolation of CS delivery in Ethiopia

The descriptive spatial analysis showed that the prevalence of CS delivery in Ethiopia was highest in Addis Abeba, and Diredewa, and low in Afar, Tigray, and Benishangul Gumz regions (**Fig 2**).

The spatial autocorrelation showed that there is a clustering of CS delivery [MI = 1.009, Z score = 4.99, and p-value <0.001] in Ethiopia (**Fig 3**).

The Hotspot analysis showed that CS delivery was hotspots [high prevalence] in Diredewa, Harerie, Somali, and Oromia regions, and cold spots [low prevalence] of CS delivery in Tigray, Amhara, Afar, and Benishangul Gumz region (**Fig 4**).

**Table 1. Socio-demographic characteristics of mothers who gave birth 5 years before the survey in Ethiopia, 2019 Mini-EDHS (weighted).**

| Variable | Category | Cesarean section delivery | | | |
|---|---|---|---|---|---|
| | | No | % | Yes | % |
| Age category | 15–24 | 124 | 52 | 51 | 48 |
| | 25–34 | 2512 | 93.8 | 165 | 6.2 |
| | 35–49 | 1199 | 6.6 | 85 | 93.4 |
| Sex of household | Male | 4538 | 95 | 237 | 5 |
| | Female | 688 | 91.6 | 63 | 8.4 |
| Place of residence | Urban | 1229 | 89.9 | 138 | 11.1 |
| | Rural | 3998 | 96.1 | 162 | 3.9 |
| Religion | Christian | 3106 | 93.1 | 232 | 6.9 |
| | Muslim | 2120 | 96.8 | 69 | 3.2 |
| Region | Tigray | 346 | 93.3 | 25 | 6.7 |
| | Afar | 84 | 96.5 | 3 | 3.5 |
| | Amhara | 972 | 92.57 | 78 | 7.43 |
| | Oromia | 2120 | 95.9 | 90 | 4.1 |
| | Somali | 404 | 99.1 | 4 | 0.98 |
| | Benishngul Gumz | 63 | 94.03 | 4 | 5.97 |
| | SNNP | 1054 | 95.38 | 51 | 4.62 |
| | Gambella | 24 | 96 | 1 | 4 |
| | Hareri | 15 | 88.24 | 2 | 11.76 |
| | Addis Abeba | 118 | 75.64 | 38 | 24.36 |
| | Diredewa | 25 | 83.3 | 5 | 16.7 |
| Marital status | Union | 2951 | 91.5 | 274 | 8.5 |
| | Not union | 276 | 91.1 | 27 | 8.9 |
| Educational status | Cannot read and write | 2875 | 97.1 | 86 | 2.9 |
| | Primary education | 1875 | 93.75 | 125 | 6.25 |
| | secondary education | 1832 | 97.23 | 52 | 2.77 |
| | Higher education | 362 | 90.5 | 38 | 9.5 |
| Wealth index | Poorest | 1300 | 98.4 | 21 | 1.6 |
| | Poorer | 1154 | 96.33 | 44 | 3.67 |
| | Middle | 997 | 95.5 | 47 | 4.5 |
| | Richer | 915 | 95.3 | 45 | 4.7 |
| | Richest | 861 | 85.8 | 144 | 14.3 |

The spatial interpolation of CS delivery for locations where data were not obtained has been estimated using the standard Kriging technique. The highest estimated prevalence of CS delivery (red shaded) was observed in Addis Ababa, Diredewa, and parts of Somalia regions (**Fig 5**).

## Prevalence of cesarean section delivery in Ethiopia

The prevalence of cesarean section delivery was 5.4% (95% CI (4.6%-6.4%)) in Ethiopia.

## Individual and community-level factors of cesarean section delivery in Ethiopia

In the final model (model-III) maternal age, sex of the household, religion, educational status, types of birth from individual-level variables, and the region from community-level variables were associated with cesarean section delivery (Table 3). The odds of CS delivery were 1.98 and 3.7 times more among mothers whose ages are in the age group 25–34 and 35–49 as compared to mothers in the age group15-24 [AOR = 1.98, 95% CI = (1.3, 3.1)] and [AOR = 3.7,

**Table 2. The maternal health service utilization of reproductive-age mothers who gave birth 5 years before the survey in Ethiopia Using 2019 Mini-EDHS (weighted).**

| Variable | Category | Cesarean section delivery | | | |
|---|---|---|---|---|---|
| | | No | % | Yes | % |
| Last delivery by previous CS | Yes | 4653 | 99.6 | 20 | 0.4 |
| | No | 36 | 13.3 | 234 | 86.7 |
| ANC Visit | No | 993 | 98.9 | 11 | 1.1 |
| | Yes | 2689 | 91.99 | 234 | 8.01 |
| Types of birth | Single | 5109 | 94.96 | 271 | 5.04 |
| | Multiple | 118 | 80.3 | 29 | 19.7 |
| Sex of New Birth | Male | 2692 | 94.7 | 150 | 5.3 |
| | Female | 2535 | 94.4 | 150 | 5.6 |
| Place of delivery | Public health facility | 2235 | 89.2 | 270 | 10.8 |
| | Private hospital | 149 | 83.2 | 30 | 16.8 |

95% CI = (2.2, 6.1)] respectively. The odds of CS delivery were 1.9 times higher among participants who lived in female head households as compared to male head households [AOR = 1.9, 95% CI = (1.2, 3.1)]. The odds of CS delivery were 60% less likely among

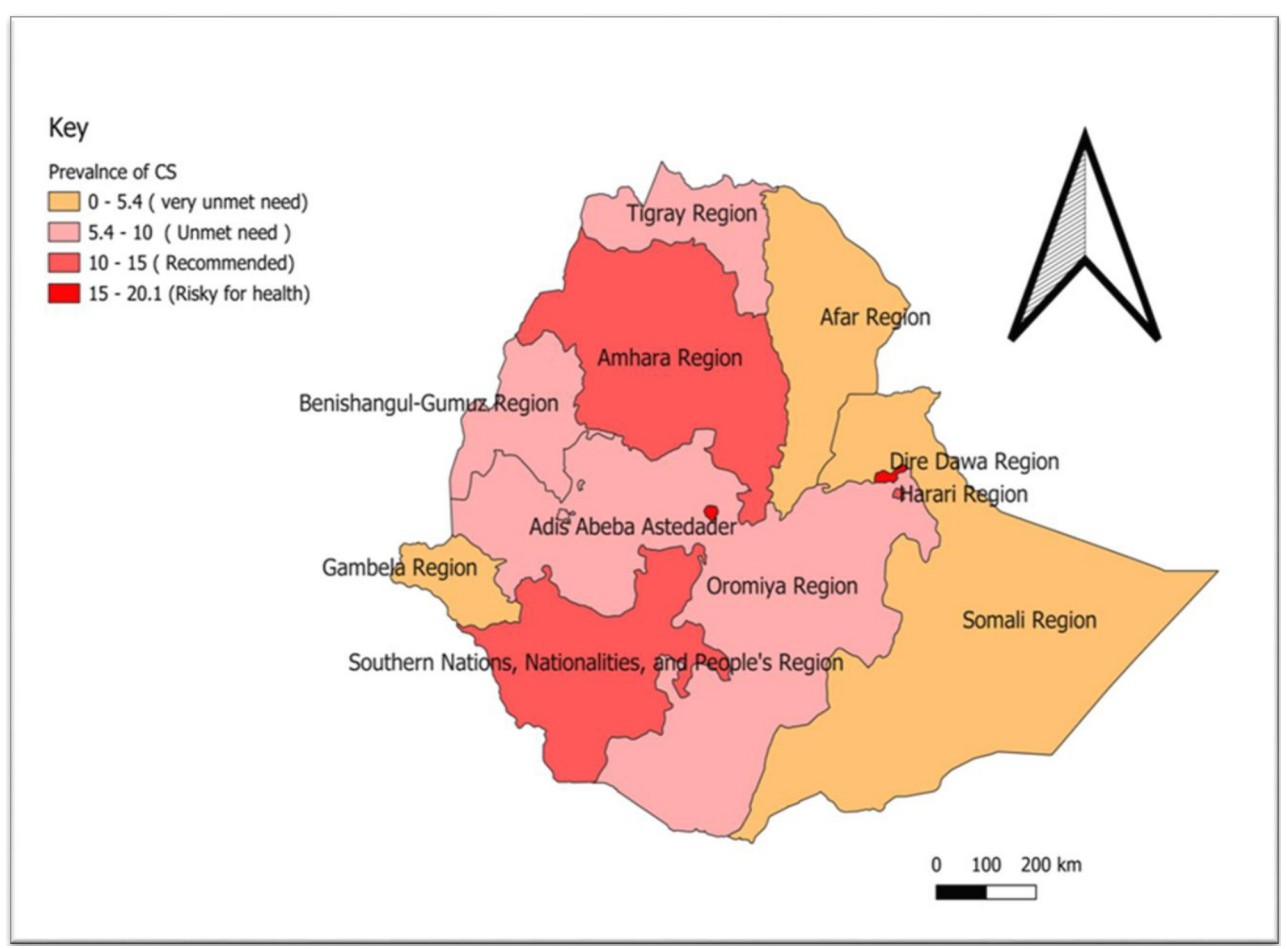

**Fig 2. Spatial distribution of CS delivery in Ethiopia.**

# Spatial Autocorrelation Report

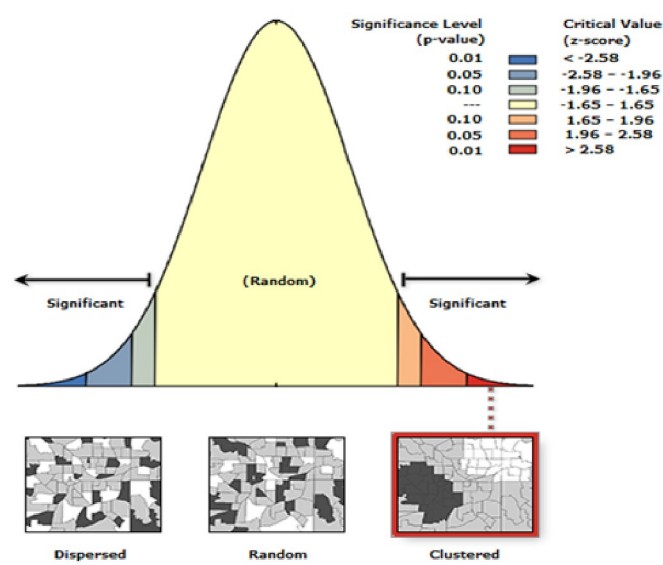

| Moran's Index: | 1.009090 |
| z-score: | 4.966507 |
| p-value: | 0.000001 |

Given the z-score of 4.9665068324, there is a less than 1% likelihood that this clustered pattern could be the result of random chance.

**Fig 3. Spatial autocorrelation analysis of CS delivery in Ethiopia.**

participants who are Muslim religion followers as compared to Christian religion followers [AOR = 0.4, 95% CI = (0.2, 0.7)]. Mothers who attended primary education, secondary education, and higher education were 1.6, 2.4, and 2.06 times more likely to have CS delivery as compared with mothers who did not attend school [AOR = 1.6, 95% CI = (01.07, 2.7)], [AOR = 2.4, 95% CI = (1.3, 4.25)], and [AOR = 2.06, 95% CI (1.03, 4.2)] respectively. The odds of CS delivery were 8.1 times higher for multiple births as compared with single births [AOR = 8.1, 95% CI = (3.4, 19.74)]. Mothers who were living in Addis Ababa and Dire Dewa were 3.6 and 7 times more likely to have cesarean section delivery as compared to mothers who live in the Tigray region [AOR = 3.4 95% CI = (1.1, 11.9)], and [AOR = 7, 95% CI = (1.23, 40.7)] respectively (Table 3).

**Random effects (measures of variation).**   Cesarean section delivery varies significantly across each cluster. ICC indicated that 44.1% of the variation in CS delivery among mothers was attributed to community-level factors. PCV in the final model showed that 29.1% of the variation in CS delivery across communities was explained. The median odds ratio confirmed that the variation in CS delivery was influenced by community-level factors. In the initial model (model zero), the median odds ratio for CS delivery was 9.2, indicating significant variation between communities (9.2 times higher than the reference or MOR = 1). However, when all factors were added to the final model, the unexplained community variation decreased to an MOR of 7.9. This suggests that even after considering all factors, the effects of clustering remain statistically significant in the full models (Table 4).

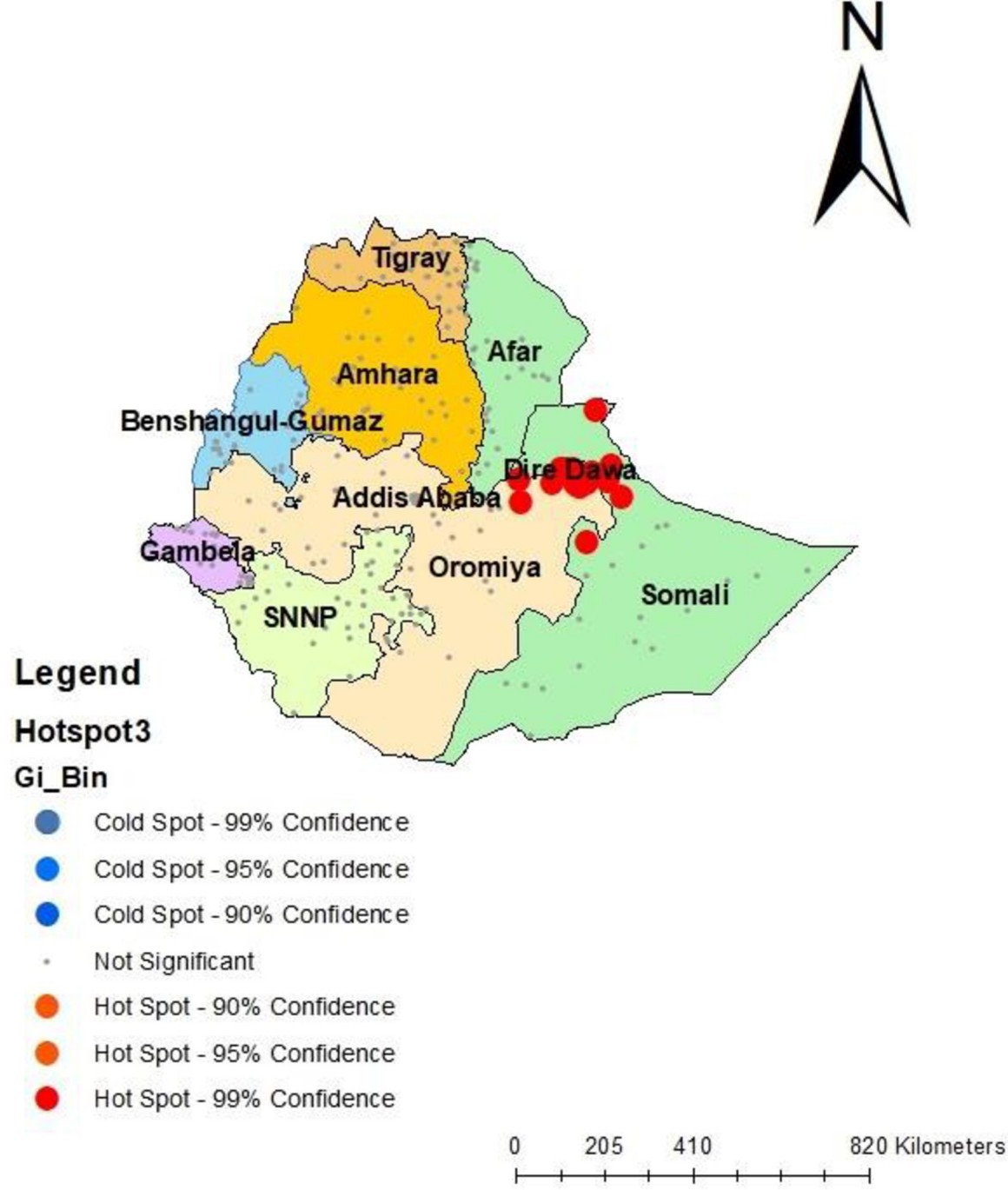

**Fig 4. Spatial hotspot analysis of CS delivery in Ethiopia.**

## Discussion

This study used the 2019 Mini-EDHS data to analyze the geospatial pattern and individual and community-level factors of CS delivery in Ethiopia. The finding of this study indicates that CS prevalence was 5.44% in Ethiopia. The findings are similar to those studies conducted in Mozambique (4.7%) [11], and Uganda (5.22%) [32]. However, this prevalence is less than the

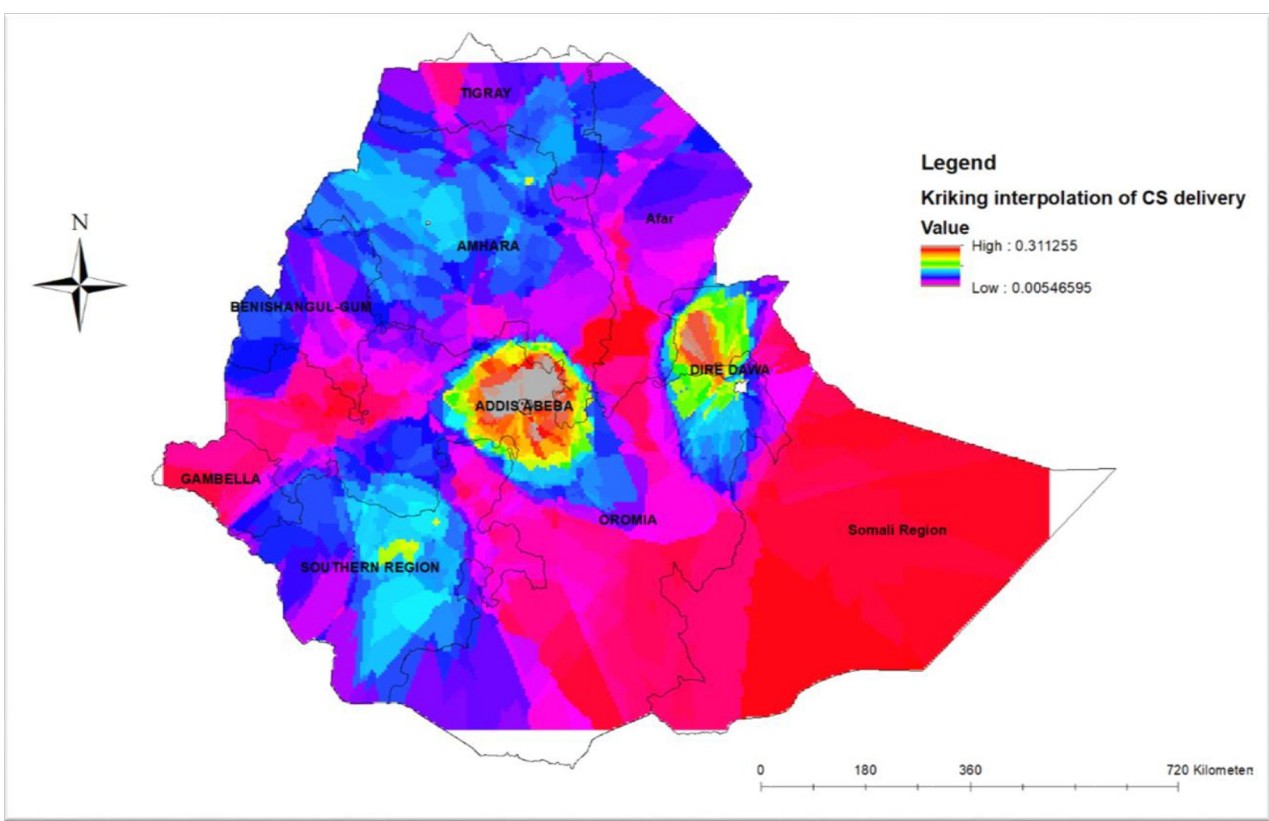

**Fig 5. Spatial interpolation analysis of CS delivery in Ethiopia.**

research conducted in Nepal (9%) [33], Egypt (40.1%) [34], Bangladesh (63%) [35], and China (37%) [36]. This is slightly increased from the previous measure EDHS report in 2011 (0.9%) [37], and 2016 (1.9%). This indicates that CS availability has slightly improved, however, it is still below the WHO recommendation which indicates unmet need as per the WHO statement [6, 7]. The rate of CS delivery and geographical disparities in the region of Ethiopia in the current study could be attributed to different factors. The higher CS delivery beyond the WHO estimates, especially in urban areas and regions with greater access to Operating Theaters and Obstetricians capable of performing the procedure could be attributed to various factors, including financial incentives for medical practitioners and convenience for both healthcare providers and clients and surged due to non-medical justifications such as financial gain, convenience, and other factors that do not align with medical necessity[4, 38]. The lower CS delivery below the WHO estimates, especially in rural and remote areas may be due to inadequate and/or sparse availability of emergency obstetrics services, inadequate equipment and medication in obstetric emergency centers, a shortage of skilled birth attendants and services, long travel distances, and unsuitable terrain that could provide a substantial geographic barrier to emergency obstetric treatment [4, 39, 40]. Related to factors, maternal age, sex of the household, religion, educational status, types of birth from individual-level variables, and, the region from community-level variables were associated with CS delivery. The CS delivery was high among participants who are in the age group 25–34 and 35–49 as compared to mothers in the age group15-24. This finding is supported by studies in Vietnam [41], Egypt [34], and Mozambique [11]. The reason might be that the risk of delivery problems as well as the chance of premature birth and infant death increased as mothers got older, excessive bleeding during labor,

**Table 3. The multilevel analysis of cesarean section delivery in Ethiopia, 2019 Mini-EDHS (weighted).**

| Variable | Category | CS delivery | | Bivariable | | Model 0 | Model 1 | Model 2 | Model 3 |
|---|---|---|---|---|---|---|---|---|---|
| | | No | Yes | P-value | 95%COR | ICC = 44% | 95%AOR | 95%AOR | 95%AOR |
| Age category | 15–24 | 124 | 51 | | Reference | | Reference | | Reference |
| | 25–34 | 2512 | 165 | 0.02 | 1.4(1.1–2) | | 1.9(1.3–3.1) | | 1.98(1.3–3.1)* |
| | 35–49 | 1199 | 85 | 0.00 | 1.7(1.2–2.5) | | 3.6(2.2–6.0) | | 3.7(2.2–6.1)* |
| Sex of household head | Male | 4538 | 237 | | Reference | | Reference | | Reference |
| | Female | 688 | 63 | 0.00 | 1.7(1.5–2.3) | | 1.8(1.1–2.9) | | 1.9(1.2–3.1)* |
| Place of residence | Urban | 1229 | 138 | | Reference | | | | Reference |
| | Rural | 3998 | 162 | 0.00 | 0.4(0.3–0.5) | | | 0.3(0.1–0.5) | 0.6(0.3–1.4) |
| Religion | Christian | 3106 | 232 | | Reference | | | | Reference |
| | Muslim | 2120 | 69 | 0.00 | 0.4(0.3–0.6) | | 0.38(0.18–0.65) | | 0.4(0.2–0.7) |
| Region | Tigray | 346 | 25 | | Reference | | Reference | Reference | Reference |
| | Afar | 84 | 3 | 0.15 | 0.4(0.1–1.5) | | | 0.4(0.1–2.3) | 2.4(0.3–23.7) |
| | Amhara | 972 | 78 | 0.74 | 1.1(0.8–1.7) | | | 1.4(0.5–3.7) | 2.4(0.8–6.6) |
| | Oromia | 2120 | 90 | 0.02 | 0.6(0.4–0.9) | | | 0.5(0.2–1.4) | 1.8(0.6–5.2) |
| | Somali | 404 | 4 | 0.00 | 0.1(0.1–0.4) | | | 0.1(0.0–0.6) | 0.6(0.05–4.3) |
| | Benshngul | 63 | 4 | 0.84 | 0.9(0.3–2.5) | | | 0.9(0.2–4.2) | 2.0(0.4–10.1) |
| | SNNP | 1054 | 51 | 0.09 | 0.7(0.4–1.0) | | | 1.0(0.4–2.7) | 1.9(0.7–5.5) |
| | Gambella | 24 | 1 | 0.56 | 0.5(0.1–4.3) | | | 0.4(0.03–5) | 0.3(0.02–6.6) |
| | Hareri | 15 | 2 | 0.83 | 1.2(0.2–7.4) | | | 1(0.1–9.6) | 2.9(0.2–38) |
| | Addis Abeba | 118 | 38 | 0.00 | 4.3(2.4–7.4) | | | 2.9(0.9–8.9) | 3.6(1.1–11.9)* |
| | Diredewa | 25 | 5 | 0.09 | 2.4(0.9–7.2) | | | 2.5(0.6–11) | 7(1.2–40.7)* |
| Marital status | Union | 2951 | 274 | | Reference | | Reference | | Reference |
| | Not union | 276 | 27 | 0.01 | 1.7(1.1–2.6) | | 1.4(0.7–2.6) | | 1.3(0.7–2.6) |
| Educational status | Cannot read and write | 2875 | 86 | | Reference | | Reference | | Reference |
| | Primary education | 1875 | 125 | 0.00 | 2.3(1.7–3.0) | | 1.7(1.1–2.6) | | 1.6(1.07–2.7)* |
| | secondary education | 1832 | 52 | 0.00 | 4.8(3.3–6.9) | | 2.3(1.3–4.1) | | 2.4(1.3–4.25)* |
| | Higher education | 362 | 38 | 0.00 | 7.9(5.2–12.6) | | 2.2(1.1–4.4) | | 2.06(1.1–4.2)* |
| Wealth index | Poorest | 1300 | 21 | | Reference | | Reference | | Reference |
| | Poorer | 1154 | 44 | 0.00 | 2.3(1.4–3.9) | | 1.4(0.7–3.1) | | 1.3(0.6–2.9) |
| | Middle | 997 | 47 | 0.00 | 2.9(1.7–4.8) | | 0.8(0.4–1.7) | | 0.7(0.3–1.6) |
| | Richer | 915 | 45 | 0.00 | 3.1(1.8–5.2) | | 0.9(0.4–1.9) | | 0.8(0.4–1.7) |
| | Richest | 861 | 144 | 0.00 | 10.4(7–17) | | 1.3(0.5–2.6) | | 0.8(0.3–1.9) |
| ANC Visit | No | 11 | 1033 | | Reference | | Reference | | Reference |
| | Yes | 272 | 2663 | 0.00 | 8.2(4.4–14.9) | | 1.9(0.8–4.3) | | 1.9(0.8–4.2) |
| Types of birth | Single | 325 | 5261 | | Reference | | Reference | | Reference |
| | Multiple | 24 | 143 | 0.00 | 4.6(3.1–7.1) | | 8(3.4–19.6) | | 8.1(3.4–19.7)* |
| Place of delivery | Public health facility | 283 | 2280 | | Reference | | Reference | | Reference |
| | Private health facility | 66 | 243 | 0.01 | 1.68(1.2–2.5) | | 1.4(0.8–2.5) | | 1.3(0.6–2.4) |

Key * = significant at 95% confidence interval, AOR = Adjusted odds ratio, COR = crude odds ratio, ICC = Intra-Class Correlation, CS = Cesarean Section,

ANC = Antenatal care, SNNP = South Nation Nationalities and Peoples

prolonged labor lasting more than 20 hours, and dysfunctional labor that does not progress to the next stage. Furthermore, diabetes and hypertension during pregnancy were more common in older pregnant mothers; this may lead to increased delivery by CS [42]. The odds of mothers who live with a female household head were more likely to deliver by CS than mothers who live with a male household head. There is no evidence to support this finding [21–23, 36, 43].

**Table 4. Model fitness, selection, and cluster variation.**

| Model | ICC | Variance | PCV | MOR | LLR | AIC | BIC |
|---|---|---|---|---|---|---|---|
| Model 0 | 44.1% | 2.3 | Reference | 9.2 | -1202 | 2408 | 2421 |
| Model 1 | | 1.67 | 27.39% | 8.01 | -639 | 1423 | 1514 |
| Model 2 | 32.07% | 1.7 | 26.09% | 8.08 | -983 | 1993 | 2079 |
| Model 3 | 31.2% | 1.63 | 29.1% | 7.9 | -632 | 1320 | 1479 |

**Key:** AIC: Akakian information Criteria, BIC: Bayesian information Criteria, ICC: Intra-class correlation, LLR = log-likelihood ratio, MOR: Median Odds Ratio, PCV: Proportional Change of Variability. The first model was the model-0 = empty model or null model was conducted without an independent variable (univariate analysis) and the result showed intra-class correlation (ICC) = 44%, the second model was model 1 = analyzing only individual-level variable, the 3rd model was model-2 (analyzing only community-level variable), the last model was model 3 = analyzing both individual and community-level variable.

Muslim religion-follower mothers were less likely to deliver by CS delivery than Orthodox religion-follower mothers. This finding is supported by the study conducted in Nigeria [10]. Mothers who were living in Addis Ababa and Dire Dewa were; more likely to have CS delivery as compared to mothers who lived in the Tigray region [44]. The odds of CS delivery among mothers who had completed primary, secondary, or higher education were more likely than mothers who had not attended education. This finding is supported by the study conducted by global, and national representatives of 150 countries using secondary data [2, 36, 41, 45–47]. The possible reason for a higher degree of education enhances the possibility of CS delivery might be mothers who have received education aware of the costs and benefits of using maternity services and also have more confidence and self-reliance for any decision provided by a healthcare provider, which in turn, may raise the chance that could get CS service, besides, evidence from health literature demonstrates the fact that health literacy is linked to educational achievement could lead to believe that highly educated mothers are more likely to seek health services such as CS delivery than their no educated counterparts [48]. And also may be associated especially in urban areas and regions with greater access to Operating Theaters and Obstetricians capable of performing the procedure, CS rates can be attributed to various factors, including financial incentives for medical practitioners and convenience for both healthcare providers and clients and non-medical justifications such as financial gain, convenience, and other factors that do not align with medical necessity [49, 50]. In this finding, multiple births (twin pregnancies) were significant factors for CS delivery. This study is supported by a study conducted in Egypt [34] and Nigeria [10]. The possible reason for this could be multiple births (twin pregnancies) have been linked to obstetric complications like preterm labor, premature rupture of membranes, malposition, and male presentation of the fetus. It also increased risks of complications to the mother like gestational diabetes, gestational hypertension, preeclampsia, and intrahepatic cholestasis, all of which may increase the likelihood of giving birth via CS delivery.

Strength and limitation: The strength of research was using nationally representative datasets, which gives sufficient statistical power and the study estimates completed after the data had been weighted to allow for probability sampling and nonresponse to make it representative at national and regional levels. The limitation of this study was some factors that affecting CS delivery, like birth weight, maternal weight and height to generate BMI were incomplete in the 2019 Mini-EDHS data.

## Conclusion

In Ethiopia, CS deliveries remain below the recommended World Health Organization rates with distinct geospatial variation. Geographically, CS prevalence is concentrated in urban

areas like Addis Ababa and Diredewa, as well as in parts of the Somali and Oromia regions. Maternal age, female household head, Muslim religion, higher educational status, and multiple births at the individual level, and region at the community level were factors of CS delivery in Ethiopia. Therefore, efforts should be geared towards tailoring strategies and interventions to ensure fair access to CS delivery lines with WHO recommendations, especially in the region where CS delivery rates are low in Tigray, Amhara, Afar, and Benishangul Gumze regions.

## Acknowledgments

The authors are sincerely grateful to the Demographic Health Survey (DHS) program for allowing us to use the Ethiopian mini DHS dataset through their archives (archive@dhspro-gram.com).

## Author Contributions

**Conceptualization:** Abel Endawkie, Ermias Bekele Enyew, Kokeb Ayele, Lakew Asmare, Fekade Demeke Bayou, Mastewal Arefaynie, Yawkal Tsega.

**Data curation:** Abel Endawkie, Mengistu Mera Mihiretu, Yawkal Tsega.

**Formal analysis:** Abel Endawkie, Shimels Derso Kebede, Yawkal Tsega.

**Investigation:** Abel Endawkie, Shimels Derso Kebede, Mengistu Mera Mihiretu.

**Methodology:** Abel Endawkie, Shimels Derso Kebede, Mengistu Mera Mihiretu, Ermias Bekele Enyew.

**Resources:** Fekade Demeke Bayou.

**Software:** Abel Endawkie.

**Supervision:** Abel Endawkie, Natnael Kebede.

**Validation:** Abel Endawkie.

**Visualization:** Abel Endawkie, Natnael Kebede.

**Writing – original draft:** Abel Endawkie, Natnael Kebede.

**Writing – review & editing:** Abel Endawkie, Natnael Kebede.

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
