## [Editor Report · Decision Letter 0]

24 Apr 2024

PONE-D-23-40362Geospatial Patterns, Prevalence, and Individual and Community Levels Factors of Cesarean Section Deliveries in Ethiopia: A Spatial and Multilevel AnalysisPLOS ONE

Dear Dr. Endawkie,

Thank you for submitting your manuscript to PLOS ONE. After careful consideration, we feel that it has merit but does not fully meet PLOS ONE’s publication criteria as it currently stands. Therefore, we invite you to submit a revised version of the manuscript that addresses the points raised during the review process.

The article is valuable and of great interest for maternal and reproductive health in Africa and the rest of the world. However it was written in non-standard English and contains numerous errors and lapses, which requires a full revision and re-writing. Please find an editor or person who fully understands and writes academic papers in order to revise the manuscript entirely. Once a fairly revised article is done, it can be submitted and contemplated again for publication in PLOS ONE. Thank you.

We look forward to receiving your revised manuscript.

Kind regards,

Alfredo Luis Fort, M.D., M.Sc., Ph.D.

Academic Editor

PLOS ONE

- https://www.researchsquare.com/article/rs-1386764/v1

- https://doi.org/10.1371/journal.pone.0282951

- https://bmcwomenshealth.biomedcentral.com/articles/10.1186/s12905-021-01484-1

In your revision ensure you cite all your sources (including your own works), and quote or rephrase any duplicated text outside the methods section. Further consideration is dependent on these concerns being addressed.

A clean copy of the edited manuscript (uploaded as the new *manuscript* file)”.

---

## [Author Response · Author response to Decision Letter 0]

1 May 2024

Authors’ Responses to Editor and Reviewers Comments

“Geospatial Patterns, Prevalence, and Individual and Community Levels Factors of Cesarean Section Deliveries in Ethiopia: A Spatial and Multilevel Analysis”

Dear PlOS One Editors and Reviewers;

We are thankful for your constructive comments. We have looked at the comments and have revised our paper accordingly. We hope our paper improved as a result of incorporating the reviewer’s and academic editor’s comments and suggestions. Here are the authors’ responses to the comments.

Please find for your kind consideration the following:

1. A revised manuscript with track changes.

2. A revised paper without tracked changes

3. A rebuttal letter that responds to each point raised by the academic editor and reviewer. 

The point by point responses of authors are written by hoping these changes would meet with your favorable consideration, we are happy to hear if there are more comments and suggestions. Please do not hesitate to let us know if you have any questions.

Yours Sincerely 

Mr. Abel Endawkie Correspondence Author 

Department of Epidemiology and Biostatistics School of Public Health College of Medicine and Health Science Wollo University Dessie Ethiopia 

Tel. 251935459310 Email address abelendawkie@gmail.com

We have tried our best to improve it accordingly:

 Please revise the manuscript.

Response 1: We revised our manuscript based on Plos one submission guidelines (manuscript, title, table, figure and reference format). 

- https://www.researchsquare.com/article/rs-1386764/v1

- https://doi.org/10.1371/journal.pone.0282951

- https://bmcwomenshealth.biomedcentral.com/articles/10.1186/s12905-021-01484-1

Response 2: We would like to thank you kindly. After careful review, we made the necessary revisions and corrections.

 Response 3: We are grateful to thank you for your kindness and response. We have tried to correct your the languge thoroughly.

Response 2: We have prepared it. 

Point-by-point response of Authors for editors' and reviewers' comment 

Editor's Comments to the Authors

Additional Editor Comments:

The authors are requested to address the reviewers' comments carefully and in detail 

Response: We are grateful to thank you for your kindness and response. We tried to correct it thoroughly. 

Dear Editors: We would like to notify you that, for more appropriate understanding of the paper we have revised the title to “Geospatial Patterns, and Individual and Community Levels Factors of Cesarean Section Deliveries in Ethiopia: A Spatial and Multilevel Analysis ”

 Response: We are grateful to thank you for your kindness and response. We have tried to correct your concern and suggestion thoroughly.

Reviewer Comment to the Authors 

Abstract section 

Comment 1 Abstract result section: this is presented differently, better to use commen words.

Response 1: We sincerely appreciate your thoughtful feedback. After careful review, we made the necessary revisions and corrections. 

Comment 2 Abstract conclusion section: The previous result no include the prevalence over time? So increase aspect is not clear please review.

Response 2: We extend our sincere gratitude for your insightful comment. We found it particularly valuable during the writing process of our work. We would like to sorry for making confuse in understanding. After careful review, we made the necessary revisions and corrections.

Comment 3: Abstract conclusion section: This sentence is not complete

Response 3: We sincerely appreciate your thoughtful comment. In response to your concern, since we found it particularly valuable during the writing process of our work and we would like to sorry for making confuse in understanding. After careful review, we made the necessary revisions and corrections.

Introduction section

Comment 4. Lesthan 5% people . It is not clear (people or pregnant women) 

Response 4: We would like to thank you kindly for your comment. We had revised and corrected it thoroughly. Our interest is to say, according to the WHO statement, CS prevalence less than 5% among reproductive age mothers in the country or specific community indicates unmet need, whereas a prevalence of CS greater than 15% among reproductive age mothers might put the mother and new birth at risk.

Comment 5: population studies it is not clear?

Response 5: We would like to thank you kindly. We found it particularly valuable during the writing process of our work and we would like to sorry for making confuse in understanding. After careful review, we made the necessary revisions and corrections.

Comment 6: what is negative if CS rate lower infant death? It is not clear 

Response 6: We express our gratitude for your insightful comment. We found it particularly valuable scientific plausibility of the paper and we would like to sorry for making confuse in understanding. After careful review, we made the necessary revisions and corrections.

Comment 7: Which WHO survey?

Response 7: First and foremost, we express our sincere gratitude for your intriguing idea and thoughtful questions. We have gained valuable insights from it. After careful review, we made the necessary revisions and corrections.

Comment 8: line 70 Explaine more.

Response 8: We sincerely appreciate your thoughtful comment. After careful review, we made the necessary revisions and corrections.

Comment 9: line 75 Start this in new paragraph given that is different topic.

Response 9: We express our sincere gratitude for your insightful comment. We have gained valuable insights from it. After careful review, we made the necessary revisions and corrections.

Comment 10. Line 77 Does it mean the rate of CS decreasing?

Response 10: We express our sincere appreciation for your insightful comment. We would like to sorry for making confuse in understanding. The rate of CS delivery in Ethiopia is increasing from 0.7 to 1.9 in 2000 to 2016 EDHS.

Commentm 11: line 80 write the numper in parenthesis after country name.

Response 11: We extend our sincere appreciation for your thoughtful comment. In response to your query, we made the necessary revisions and corrections..

Comment 12: introduce the mini EDHS data.

Response 12: We express our sincere appreciation for your insightful comment. Following your feedback, we have made revisions and incorporated your comment in manuscript.

Method section 

Comment13: you have to explaine it first MEDHS?

Response 13: First and foremost, we express our sincere gratitude for your intriguing idea and thoughtful questions. In response to your query, we have made revisions and incorporated your comment in manuscript.

 Comment 14: please explain more your sampling and sample size determination.

Response 14: We would like to thank you kindly for your deep comment. In response to your query, we have made revisions and incorporated your comment in manuscript. Please look the sampling procedure depiceted in figure 1 in detail 

Comment 15. Line 120: no need of parenthesis 

Response 15: We sincerely appreciate your thoughtful comment. The issue you raised is indeed intriguing. we have made revisions and incorporated your comment in manuscript.

Comment 16. Line 61: In which table please spacify 

Response 16: We sincerely appreciate your thoughtful comment. We provided in table 2 and 3 Please look at table a 2 and 3

Comment 17: Line 131 please explaine more types of spatial statistics 

Response 17: We sincerely appreciate your thoughtful comment. We have carefully reviewed and made the necessary corrections. 

Comment 18. Please write the equation in new line 

Response 18: We sincerely appreciate your thoughtful comment. We have carefully reviewed and made the necessary corrections.

Comment 19. Standar error never more than ±2

Response 19: We sincerely appreciate your thoughtful comment. Multicolinarity in logistic analysis is not estimated using variance inflaction factor (VIF). In this we consider the standard error of the variable. If the standard is below -2 and above 2 it indicates multiconiarity but it is does not mean that standard error never less than -2 or greater 2.

Result Section 

Comment 20. Please edit more and please specify the delivery year before the DHS survey.

Response 20: We sincerely appreciate your thoughtful comment. We have carefully reviewed and made the necessary corrections.

Discusion Section 

Comment 21. What was the previous CS delivery in Ethiopia please write 

Response 21: We would like to thank you kindly for your deep comment. We have carefully reviewed and made the necessary corrections.

We feel all of you may understand our feelings and interests.

We are thankful for all!!

---

## [Decision Letter · Decision Letter 1]

23 May 2024

PONE-D-23-40362R1Geospatial Patterns, and Individual and Community Levels Factors of Cesarean Section Deliveries in Ethiopia: A Spatial and Multilevel AnalysisPLOS ONE

Dear Dr. Endawkie,

Thank you for submitting your manuscript to PLOS ONE. After careful consideration, we feel that it has merit but does not fully meet PLOS ONE’s publication criteria as it currently stands. Therefore, we invite you to submit a revised version of the manuscript that addresses the points raised during the review process (some comments you have already addressed). 

Please check in particular for a comment about how CS have increased way beyond the WHO estimates, in particular in urban areas and in some parts of the world where there is more access to Operating Theaters and Obstetricians who can perform the operation. It is known that both the medical practitioners as well as the clients themselves decide to perform it for financial advantage, convenience, etc., and other factors that are not medically justified. Please review the literature and include this aspect. Please also see a few minor comments in the file attached. Thanks.

We look forward to receiving your revised manuscript.

Kind regards,

Alfredo Luis Fort, M.D., M.Sc., Ph.D.

Academic Editor

PLOS ONE

Journal Requirements:

Reviewers' comments:

Reviewer's Responses to Questions

**Comments to the Author**

1. If the authors have adequately addressed your comments raised in a previous round of review and you feel that this manuscript is now acceptable for publication, you may indicate that here to bypass the “Comments to the Author” section, enter your conflict of interest statement in the “Confidential to Editor” section, and submit your "Accept" recommendation.

Reviewer #1: (No Response)

Reviewer #2: (No Response)

2. Is the manuscript technically sound, and do the data support the conclusions?

Reviewer #1: Partly

Reviewer #2: Yes

3. Has the statistical analysis been performed appropriately and rigorously? 

Reviewer #1: No

Reviewer #2: Yes

4. Have the authors made all data underlying the findings in their manuscript fully available?

Reviewer #1: No

Reviewer #2: Yes

5. Is the manuscript presented in an intelligible fashion and written in standard English?

Reviewer #1: No

Reviewer #2: No

6. Review Comments to the Author

Reviewer #1: Overall I appreciate your work that you try to assess the geospatial, individual and community factors that affect CS deliveries in Ethiopia using the recent (2019) mini EDHS. It is very interested. But I have some comments and suggestions below

1. In line 13, there is numeric number 4, what does that mean?? if it is an editing problem remove it

2. In your abstract of background section(from line 22-25) , the paragraph is not well written, as the abstract is the blue print of the whole document, it should describe document shortly and contain the introduction, the statement of problem specially the magnitude and severity of problem and significance of the study clearly in one paragraph. please rewrite it again

3. In abstract of methods section from line 28-29, which model did you use to identify the hotspot and cold spot areas and explore the spatial distribution of CS in Ethiopia? And which types of multilevel analysis model did you use to identify factors affecting CS in Ethiopia?

4. Line 31, you try to figure the prevalence of CS in Ethiopia, but in your title I didn’t see the prevalence of CS in Ethiopia?? Why?

5. Line 53, not start with abbreviation after(.) if you want to use it write it as full phrases

6. Line 53-55 you sated that CS is done for maternal and fetal reasons and it is more frequent than before and the trend is expected to continue in the next ten years, WHY??

7. Line 55, reference?

8. There is the redundancy e.g from line 62-65 there is the prevalence is used three times in one paragraph , please paraphrase it

9. Line 68, where this population based study was conducted??

10. Line 71, if you use the abbreviation CS omit the word section next to it

11. Line 74 either maternal health or health of mother is enough, so remove one

12. Once you abbreviate using the full phrase, abbreviating it using same fashion is not necessary, means, in line 66 you abbreviate the EDHS with its full phrase and repeate it in line 85-86.

13. Generally, your introduction part is not consistent or well written, meaning in some paragraphs you talk about the necessity of CS and its importance, in other paragraphs you talk about its complication. this two ideas will contradicted.

14. In your result section use table format

Reviewer #2: Dear Author,

I commend you for your work, which is substantial. However, the manuscript is not well organized, and it looks as if the version submitted is a draft:

1. There are unnecessary line numbering that need to be removed across the manuscript.

2. The headlines and titles are not numbered in a logical sequence.

3. There are crossed sentences (in red) from previous reviewers with corrections not applied.

7. PLOS authors have the option to publish the peer review history of their article (what does this mean?). If published, this will include your full peer review and any attached files.

Reviewer #1: No

Reviewer #2: **Yes: **Dr. AMOS MAKELELE M'YISA

---

## [Author Response · Author response to Decision Letter 1]

8 Jun 2024

Authors’ Responses to Editor and Reviewers Comments

“Geospatial Patterns, and Individual and Community Levels Factors of Cesarean Section Deliveries in Ethiopia: A Spatial and Multilevel Analysis”

Dear Plos One Editors and Reviewers;

We are thankful for your constructive comments. We have looked at the comments and have revised our paper accordingly. We hope our paper improved as a result of incorporating the reviewer’s and academic editor’s comments and suggestions. Here are the authors’ responses to the comments.

Please find for your kind consideration the following:

1. A revised manuscript without track changes.

2. A revised paper with tracked changes

3. A rebuttal letter that responds to each point raised by the academic editor and reviewer. 

The point-by-point responses of authors are written by hoping these changes would meet with your favorable consideration, we are happy to hear if there are more comments and suggestions. Please do not hesitate to let us know if you have any questions.

Yours Sincerely 

Mr. Abel Endawkie Correspondence Author 

Department of Epidemiology and Biostatistics School of Public Health College of Medicine and Health Science Wollo University Dessie Ethiopia 

Tel. 251935459310 Email address abelendawkie@gmail.com

We have tried our best to improve it accordingly:

Point-by-point response of Authors for editor's and reviewer's comment 

Editors 

Thank you for submitting your manuscript to PLOS ONE. After careful consideration, we feel that it has merit but does not fully meet PLOS ONE’s publication criteria as it currently stands. Therefore, we invite you to submit a revised version of the manuscript that addresses the points raised during the review process (some comments you have already addressed). 

Editor comment: Please check in particular for a comment about how CS has increased way beyond the WHO estimates, in particular in urban areas and in some parts of the world where there is more access to Operating Theaters and Obstetricians who can perform the operation. It is known that both the medical practitioners as well as the clients themselves decide to perform it for financial advantage, convenience, etc., and other factors that are not medically justified. Please review the literature and include this aspect. Please also see a few minor comments in the file attached. 

Response: We are grateful to thank you for your kindness, response, insight, and suggestion. Let’s delve into why CS has increased way beyond the WHO estimates, in particular in urban areas and in some parts of the world where there is more access to Operating Theaters and Obstetricians who can operate. The cesarean sections (CS) have increased significantly beyond the WHO estimates, especially in urban areas and regions with greater access to Operating Theaters and Obstetricians capable of performing the procedure could be attributed to various factors, including financial incentives for medical practitioners and convenience for both healthcare providers and clients. Sometimes may be due to non-medical justifications such as financial gain, convenience, and other factors that do not align with medical necessity. Dear editor we incorporate this literature in our manuscript and specialy we cite in the discussion sec tion of our finding. Thank you a lot.

Reference:

1. Boerma, T., Ronsmans, C., Melesse, D. Y., Barros, A. J., Barros, F. C., Juan, L., ... & Temmerman, M. (2018). Global epidemiology of use of and disparities in caesarean sections. The Lancet, 392(10155), 1341-1348.

2. Betrán, A. P., Ye, J., Moller, A. B., Zhang, J., Gülmezoglu, A. M., & Torloni, M. R. (2016). The increasing trend in caesarean section rates: global, regional and national estimates: 1990-2014. PloS One, 11(2), e0148343.

3. Vogel, J. P., Betrán, A. P., Vindevoghel, N., Souza, J. P., Torloni, M. R., Zhang, J., ... & Temmerman, M. (2015). Use of the Robson classification to assess caesarean section trends in 21 countries: a secondary analysis of two WHO multicountry surveys. The Lancet Global Health, 3(5), e260-e270.

4. Molina, G., Weiser, T. G., Lipsitz, S. R., Esquivel, M. M., Uribe-Leitz, T., Azad, T., ... & Gawande, A. A. (2015). Relationship between cesarean delivery rate and maternal and neonatal mortality. JAMA, 314(21), 2263-2270.

Reviewers comment 

Reviewer #1 comment

General comment: Overall I appreciate your work that you try to assess the geospatial, individual and community factors that affect CS deliveries in Ethiopia using the recent (2019) mini EDHS. It is very interested. But I have some comments and suggestions below.

Response: We are grateful to thank you for your kindness and response. We have tried to correct your insightful comment thoroughly.

Comment 1: In line 13, there is numeric number 4, what does that mean?? if it is an editing problem remove it

Response 1: We are grateful to thank you for your kindness and response. This is improper placement to show author affiliation order. After careful review, we made the necessary revisions and corrections.

comment 2: In your abstract of background section(from line 22-25) , the paragraph is not well written, as the abstract is the blue print of the whole document, it should describe document shortly and contain the introduction, the statement of problem specially the magnitude and severity of problem and significance of the study clearly in one paragraph. please rewrite it again

Response 2: We extend our sincere gratitude for your insightful comment. After careful review, we made the necessary revisions and corrections. We rewrite the as” Cesarean Section (CS) is the most popular surgery worldwide in obstetric care to save a mother's or the fetus's life. The prevalence of CS delivery in Ethiopia was 0.7% and 1.9% in 2000 and 2016 respectively and its spatial distribution and variation in Ethiopia are limited. This study provides evidence for healthcare providers and pregnant women on the national CS geospatial distribution and variation to promote evidence-based decision-making and improve maternal and neonatal outcomes. Therefore, this study aimed to determine geospatial patterns and individual and community-level factors of CS deliveries in Ethiopia”.

Comment 3: In abstract of methods section from line 28-29, which model did you use to identify the hotspot and cold spot areas and explore the spatial distribution of CS in Ethiopia? And which types of multilevel analysis model did you use to identify factors affecting CS in Ethiopia?

Response 3: We extend our sincere gratitude for your insightful comment. We used the spatial hotspot analysis using Getis-Ord Gi* hot spot analysis of ArcGIS version 10.7.1 to examine where spatial clusters of CS delivery occur on the landscape by identifying areas with statistically significant clustering of relatively high values (hot spots) or relatively low values (cold spots).

Comment 4: Line 31, you try to figure the prevalence of CS in Ethiopia, but in your title I didn’t see the prevalence of CS in Ethiopia?? Why?

Response 4: We extend our sincere gratitude for your insightful comment. We understand and value your comment. To make short our title and the usual work, we believe the spatial pattern indicates the descriptive spatial distribution which introduces the prevalence of CS and spatial variation of CS delivery in Ethiopia. We appreciate your concern and suggestion.

Comment 5: Line 53, not start with abbreviation after(.) if you want to use it write it as full phrases

Response 5: We extend our sincere gratitude for your insightful comment. After careful review, we made the necessary revisions and corrections.

Comment 6: Line 53-55 you sated that CS is done for maternal and fetal reasons and it is more frequent than before and the trend is expected to continue in the next ten years, WHY??

Response 6: We extend our sincere gratitude for your insightful comment. Even though there is still a significant lack of safe CS in resource-constrained nations, the possible reason is particularly in urban areas and in some parts of the world, cesarean sections (CS) will be increased significantly beyond the WHO estimates, especially in urban areas and regions with greater access to Operating Theaters and Obstetricians capable of performing the procedure(1). The rise in CS rates will be attributed to various factors, including financial incentives for medical practitioners and convenience for both healthcare providers and clients. In some parts of the world, CS rates have surged due to non-medical justifications such as financial gain, convenience, and other factors that do not align with medical necessity. 

Beyond this there number of reasons listed in the literature.

1. Increasing maternal age: As more women are delaying childbirth to later in life, there is a higher likelihood of complications during pregnancy and delivery, leading to a higher rate of cesarean sections for maternal and fetal reasons.

2. Rising rates of obesity: The global increase in obesity rates has been associated with a higher risk of pregnancy complications, such as gestational diabetes and hypertension, which may necessitate the use of cesarean sections for the health and safety of both the mother and the baby.

3. Medical advancements: Advances in medical technology and techniques have made cesarean sections safer and more accessible, leading to an increase in the frequency of the procedure for both maternal and fetal indications.

4. Changing attitudes towards childbirth: There is a shift in societal norms and preferences towards elective cesarean sections for reasons such as convenience, fear of labor pain, or perceived control over the birthing process, contributing to the overall increase in cesarean section rates 

5. Legal considerations: Concerns over potential litigation in cases of adverse outcomes during vaginal delivery may lead healthcare providers to opt for cesarean sections more frequently, particularly when there are perceived risks to the health of the mother or the baby.

Comment 7: Line 55, reference?

Response 7: We extend our sincere gratitude for your insightful comment. After careful review, we made the necessary revisions and corrections.

Comment 8: There is the redundancy e.g from line 62-65 there is the prevalence is used three times in one paragraph , please paraphrase it

Response 8: We extend our sincere gratitude for your insightful comment. After careful review, we made the necessary revisions and corrections.

Comment 9: Line 68, where this population based study was conducted??

Response 9: We extend our sincere gratitude for your insightful comment. Dear reviewer stating the specific area is more informative for the reader. However, the reason why we were not writing was due to we believed there was a reference. Since we believe it is a valuable and reasonable suggestion we made the necessary revisions and corrections. A worldwide population‐based ecological study………

Comment 10: Line 71, if you use the abbreviation CS omit the word section next to it

Response 10: We extend our sincere gratitude for your insightful comment. After careful review, we made the necessary revisions and corrections.

Comment 11. Line 74 either maternal health or health of mother is enough, so remove one

Response 11: We extend our sincere gratitude for your insightful comment. After careful review, we made the necessary revisions and corrections.

Comment 12. Once you abbreviate using the full phrase, abbreviating it using same fashion is not necessary, means, in line 66 you abbreviate the EDHS with its full phrase and repeate it in line 85-86.

Response 12: We extend our sincere gratitude for your insightful comment. After careful review, we made the necessary revisions and corrections

Comment 13: Generally, your introduction part is not consistent or well written, meaning in some paragraphs you talk about the necessity of CS and its importance, in other paragraphs you talk about its complication. This two ideas will contradicted.

Response 13: We extend our sincere gratitude for your insightful comment. After careful review, we made the necessary revisions and corrections. Generally, CS is important to save life of the infant and mother. However, if it is surged due to non-medical justifications such as financial gain, convenience, and other factors that do not align with medical necessity will end up with maternal health problems. Therefore this study emphasizes the need for prodding evidence for healthcare providers and pregnant women on the national CS geospatial patterns, and individual and community level factors to promote evidence-based decision-making and improve maternal and neonatal outcomes. We would like to provide evidence for the police maker.

The higher CS delivery beyond the WHO estimates, especially in urban areas and regions with greater access to Operating Theaters and Obstetricians capable of performing the procedure could be attributed to various factors, including financial incentives for medical practitioners and convenience for both healthcare providers and clients and surged due to non-medical justifications such as financial gain, convenience, and other factors that do not align with medical necessity [4, 38]. However, lower CS delivery” below the WHO estimates”, especially in rural and remote areas may be due to inadequate and/or sparse availability of emergency obstetrics services, inadequate equipment and medication in obstetric emergency centers, a shortage of skilled birth attendants and services, long travel distances, and unsuitable terrain that could provide a substantial geographic barrier to emergency obstetric treatment [4, 39, 40]”. Therefore, efforts should be geared towards strategies and interventions to ensure fair access to CS delivery in line with WHO recommendations, especially in the regions where the CS delivery rate is below WHO estimates in Tigray, Amhara, Afar, and Benishangul Gumze regions

Comment 14. In your result section use table format

Response 14: We extend our sincere gratitude for your insightful comment. After careful review, we made the necessary revisions and corrections.

Reviewer #2 comment: 

Dear Author, I commend you for your work, which is substantial. However, the manuscript is not well organized, and it looks as if the version submitted is a draft:

Response: We extend our sincere gratitude for your insightful comment. After careful review, we made the necessary revisions and corrections.

Comment 1. There are unnecessary line numbering that need to be removed across the manuscript.

Response 1: We extend our sincere gratitude for your insightful comment. After careful review, we made the necessary revisions and corrections.

Comment 2. The headlines and titles are not numbered in a logical sequence.

Response 2: We extend our sincere gratitude for your insightful comment. After careful review, we made the necessary revisions and corrections.

Comment 3. There are crossed sentences (in red) from previous reviewers with corrections not applied(I have previously reminded you to look into the literature because in urban areas, with more educated mothers, they often request to be delivered by CS, plus there is an issue of more non-medically indicated CS by doctors in areas of higher socioeconomic conditions)

Response 3: We extend our sincere gratitude for your insightful comment. After careful review, we made the necessary revisions and corrections. I cited the literature indicating your query. CS delivery also may be associated especially in urban areas and regions with greater access to Operating Theaters and Obstetricians capable of performing the procedure, CS rates can be attributed to various factors, including financial incentives for medical practitioners and convenience for both healthcare providers and clients and non-medical justifications such as financial gain, convenience, and other factors that do not align with medical necessity [46, 47]. 

We appreciate your valuable time and effort in reviewing our work!!

---

## [Editor Report · Decision Letter 2]

11 Jun 2024

Geospatial Patterns, and Individual and Community Levels Factors of Cesarean Section Deliveries in Ethiopia: A Spatial and Multilevel Analysis

PONE-D-23-40362R2

Dear Dr. Endawkie,

We’re pleased to inform you that your manuscript has been judged scientifically suitable for publication and will be formally accepted for publication once it meets all outstanding technical requirements.

Kind regards,

Alfredo Luis Fort, M.D., M.Sc., Ph.D.

Academic Editor

PLOS ONE

Additional Editor Comments (optional):

You have adequately addressed recommendations by reviewers. However, there are still very minor amendments/edits (some of which are in my attached file), which can be done at the publication level. Good effort.

---

## [Editor Report · Acceptance letter]

18 Jun 2024

PONE-D-23-40362R2 

PLOS ONE

Dear Dr. Endawkie, 

I'm pleased to inform you that your manuscript has been deemed suitable for publication in PLOS ONE. Congratulations! Your manuscript is now being handed over to our production team.

Kind regards, 

on behalf of

Dr. Alfredo Luis Fort 

Academic Editor

PLOS ONE